# Quality-Switched Nd:YAG 1064 nm Laser for Management of Hyperpigmentation and Atrophic Scars after Long-Pulsed Nd:YAG Laser for Treatment of Leg Telangiectasias—A Case Report

Kristine Heidemeyer *[ID], S. Morteza Seyed Jafari, Maurice A. Adatto, Laurence Feldmeyer, Nikhil Yawalkar [ID] and Simon Bossart [ID]

Department of Dermatology, Inselspital, Bern University Hospital, University of Bern, 3010 Bern, Switzerland; madatto@skinpulse.ch (M.A.A.); laurence.feldmeyer@insel.ch (L.F.); nikhil.yawalkar@insel.ch (N.Y.); simon.bossart@insel.ch (S.B.)
* Correspondence: kristine.heidemeyer@insel.ch; Tel.: +41-31-632-2218; Fax: +41-31-632-22-33

**Abstract:** The correction of leg telangiectasias is one of the most frequently performed interventions in the Western world. While sclerotherapy remains the gold standard of treatment, several studies have shown comparable efficacy and, in some situations, an even more favorable use of lasers as an alternative treatment option. The most frequent side effect of both treatment options is hyperpigmentation, which usually clears spontaneously in most cases but can be challenging to treat if it persists. The origin of this hyperpigmentation is not fully understood; small studies point to hemosiderin as the causative pigment, at least in post-sclerotherapy hyperpigmentation. More rare side effects of the treatment include ulcerations and scarring. Quality-switched (QS) Nd:YAG lasers have demonstrated good efficacy in treating hemosiderin depositions in the skin, post-inflammatory hyperpigmentation and atrophic scars. We present a case of post-inflammatory hyperpigmentation and scarring after laser treatment of leg telangiectasia with a long-pulsed Nd:YAG laser that was successfully treated using a QS Nd:YAG 1064 nm laser. This case suggests the QS Nd:YAG laser as a possible treatment option in cases of hyperpigmentation with various origins, including hemosiderin and melanin, and scarring after laser treatment of leg telangiectasias.

**Keywords:** post-sclerotherapy hyperpigmentation; telangiectasias; veins; QS laser; 1064 nm; hemosiderin; pigment laser

## 1. Introduction

Leg telangiectasias are highly prevalent and are estimated to affect over 80% of the population [1]. Sclerotherapy is widely acknowledged as the gold standard therapy for this condition [2,3]. Among the potential side effects of sclerotherapy, hyperpigmentation stands out, with a prevalence ranging from 2% to 73%, depending on the sclerosant type and concentration; approximately 7.5% of cases exhibit persistence [2]. Long-pulsed neodymium-doped yttrium aluminum garnet (Nd:YAG) 1064 nm lasers have been suggested as an alternative treatment option and have demonstrated similar or even better efficacy in treating leg telangiectasias <1 mm [3]. However, side effects, such as ulcerations healing with scarring and hyperpigmentation, can also occur after laser treatment [3–5]. To our knowledge, the origin of hyperpigmentation after the laser treatment of leg telangiectasias has never been studied in detail.

From a histopathological study involving six patients with hyperpigmentation after classic sclerotherapy, we know that hemosiderin, and not melanin, is the main causative pigment in this context [6]. The Quality-switched (QS) Nd:YAG 1064 nm laser efficiently removes exogenous and endogenous pigment [7]. The rapid heating of the pigment leads

to its thermal expansion, causing fragmentation and the release of an acoustic wave, which damages pigment particles in the vicinity [7,8]. Fragmented pigment particles can be more easily phagocytosed and transported away by the immune system [7,8]. The QS Nd:YAG 1064 nm laser has been successfully used to treat post-inflammatory hyperpigmentation after acne, hyperpigmented keloids and axillary hyperpigmentation [9,10]. Other QS lasers, such as the QS ruby 694 nm laser or the alexandrite 755 nm laser, have been able to treat hyperpigmentation after sclerotherapy in isolated cases [11,12]. Additionally, the QS Nd:YAG laser can significantly improve the appearance of atrophic scars by stimulating collagen synthesis and remodeling, as demonstrated in a study investigating its effects on facial acne scars [13].

In this context, we present a case involving hyperpigmentation and atrophic scars following the treatment of leg telangiectasias with a long-pulsed Nd:YAG laser that was successfully managed using a QS Nd:YAG 1064 nm laser.

## 2. Case Report

A 52-year-old female patient presented to the laser unit of the Department of Dermatology, University Hospital of Bern, with hyperpigmentation and atrophic scars on her lower legs and knees following treatment for leg telangiectasias. She had undergone several sessions of sclerotherapy, and two months prior to her visit, she had received two sessions of long-pulsed Nd:YAG 1064 nm laser treatment. The laser treatment elicited a strong reaction, resulting in blister formation. Upon initial assessment, numerous dark brown hyperpigmented spots were observed, accompanied by multiple lesions with atrophy and fibrosis on both legs (Figure 1).

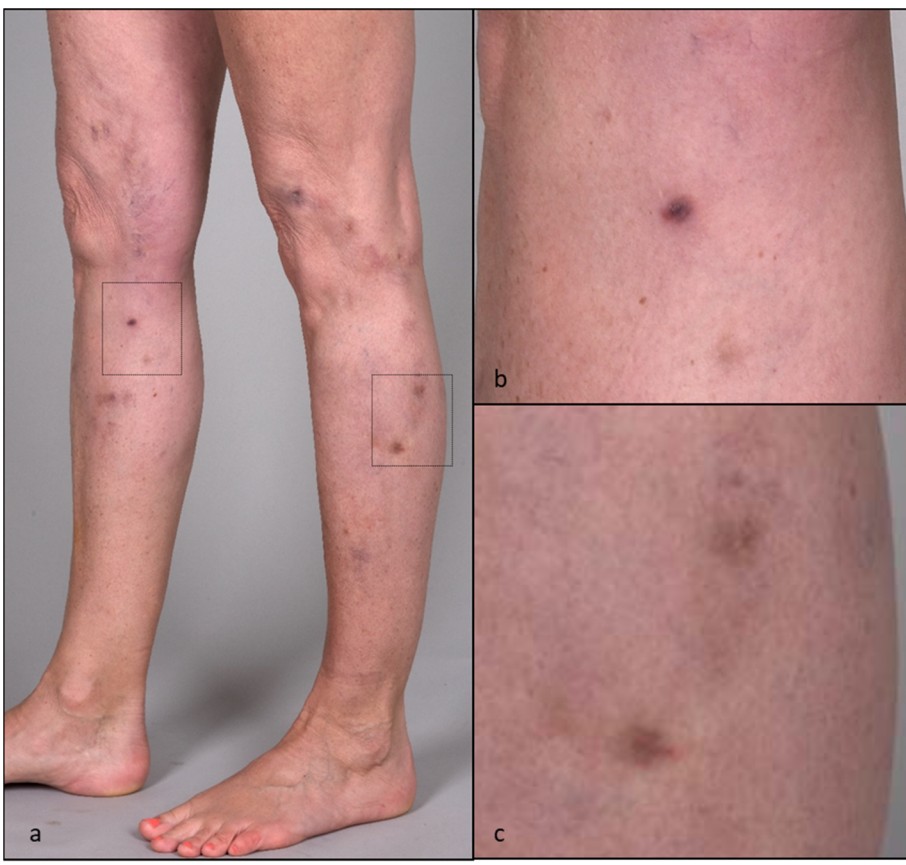

**Figure 1.** Clinical presentation of hyperpigmentation and atrophic scar on the knees and lower legs: (**a**) overview of the legs, (**b**) close up of the medial right knee, and (**c**) close up of the lateral left lower leg.

We prescribed an intensive anti-inflammatory therapy using clobetasone propionate ointment for one month to prevent the progression of post-inflammatory hyperpigmentation (PIH). Subsequent follow-up appointments revealed stabilized findings without either worsening or improvement. Consequently, a treatment plan involving a bleaching cream containing hydroquinone, tretinoin and dexamethasone (Pigmanorm®), in conjunction with sun protection, was initiated for 3 months. However, this treatment approach led to irritation. By the five-month mark post-treatment, persistent PIH was still evident in certain lesions, while another hyperpigmented area on the right knee was fading and leaving behind an atrophic scar.

For the persistent PIH, a test spot was treated with the QS Nd:YAG laser (Medlite C6, Cynosure, Westford, MA, USA). The test spot demonstrated a moderate improvement in pigmentation. Based on these results, four additional sessions of QS Nd:YAG laser 1064 nm (spot size 4 mm, fluence 6.6–6.7 J/cm$^2$) were administered on the hyperpigmented spots and scars at intervals of 2–4 months. Notably, four months after the final session, the hyperpigmentation had completely resolved, and a flattening of the atrophic scar was observed (Figure 2a–f).

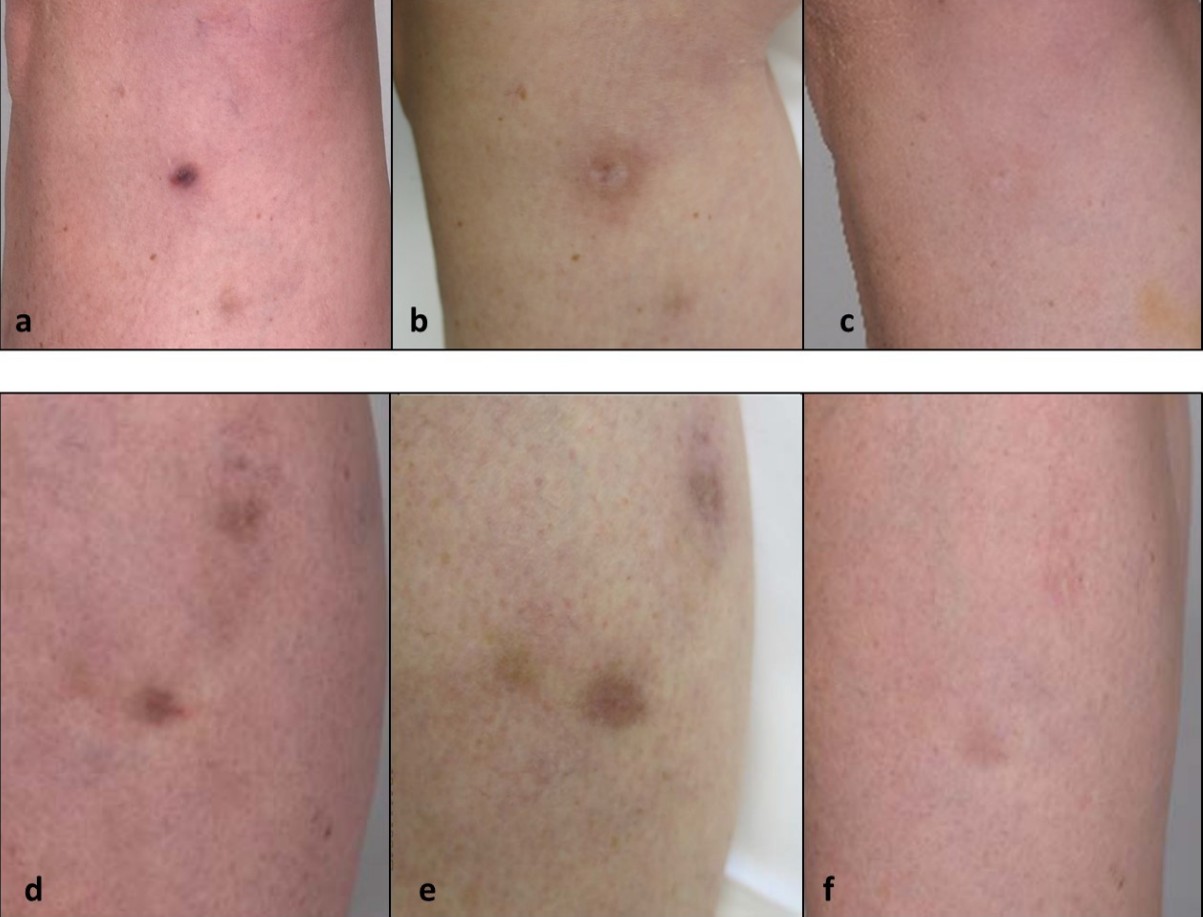

**Figure 2.** Clinical presentation of hyperpigmentation and atrophic scar before and after treatment (**a**) on the right knee before treatment, (**b**) improvement of pigment but persistence of scar after 1 month of clobetasol propionate ointment and 3 months of bleaching cream treatment, (**c**) improvement of the scar and complete resolution of the pigment on the right knee after five treatment sessions with the QS Nd:YAG 1064 nm laser, (**d**) hyperpigmentation on the left lower leg before treatment, (**e**) lack of improvement after 1 month of clobetasol propionate ointment and 3 months of bleaching cream, and (**f**) clearance of pigment after five treatment sessions QS Nd:YAG 1064 nm laser.

### 3. Discussion

The long-pulsed Nd:YAG laser represents a promising alternative for treating leg veins, especially those measuring <1 mm. This includes matting, needle-phobic patients, telangiectasias resistant to sclerotherapy, and individuals intolerant to sclerosing agents [1,3,14]. Among the side effects encountered in both sclerotherapy and laser treatment, hyperpigmentation is the most common, with a prevalence of 7.6% for the latter [1,2]. In post-sclerotherapy hyperpigmentation, over 80% of cases spontaneously clear within 1–2 years [2,6], but rarer complications, such as ulcerations and scarring, have been linked to stacked pulses, excessive treatment and inappropriate fluence adaptation [3].

While the precise cause of hyperpigmentation following sclerotherapy and laser treatment of leg veins remains unclear, evidence supports the role of hemosiderin. This stems from the disruption of vessel walls during treatment, leading to the extravasation of blood cells and contributing to pigmentation [2,3,6]. However, this understanding is primarily based on a histopathological examination of just six patients [6]. Larger studies are needed to rule out other potential causes, such as melanin incontinence from post-inflammatory hyperpigmentation, which could also play a role in both post-sclerotherapy and post-laser hyperpigmentation. Given that post-inflammatory hyperpigmentation is a prevalent side effect of laser treatments, this hypothesis requires thorough investigation.

Our patient's case provides some insight. Clearance of pigment in one hyperpigmented area after bleaching cream treatment, with no change in pigmentation in other spots even after a year of follow-up, suggests that hemosiderin—typically unresponsive to bleaching cream—may be at least partly responsible for the pigmentation. However, the response of one spot to the bleaching cream does not definitively prove the coexistence of melanin, as spontaneous resolution in that area remains possible.

The QS Nd:YAG 1064 nm laser has effectively treated cutaneous siderosis [7]. Considering that iron and hemosiderin exhibit peak absorption around 410–415 nm and an additional absorption band around 694 nm, using shorter wavelengths could be more effective [6,7,12]. Previous case reports have indicated successful hyperpigmentation treatment after classic sclerotherapy using shorter wavelengths, namely the QS ruby 694 nm and QS alexandrite 755 nm lasers. However, opting for a 1064 nm wavelength offers the advantages of deeper penetration, enhanced safety for tanned or darker skin types, and reduced risk of post-inflammatory hyperpigmentation (PIH) [7,15]. Additionally, the QS 1064 nm laser has effectively treated post-inflammatory hyperpigmentation, even though bleaching creams are often recommended as a first-line therapy. Given the unclear origin of hyperpigmentation following laser therapy for leg telangiectasias, we chose a longer wavelength to target potential deep dermal hemosiderin while also minimizing the risk of exacerbating hyperpigmentation due to melanin stimulation, as seen in classic post-inflammatory hyperpigmentation after laser treatments—especially when using shorter wavelengths.

We adopted a significantly longer treatment interval compared to other studies involving classic PIH [10]. This approach provided ample time for the immune system to eliminate the fragmented pigment, thereby minimizing the risk of complications, particularly PIH. It also allowed us to halt treatment before planned sun exposure.

In our patient, treatment with the QS Nd:YAG 1064 nm laser not only improved hyperpigmentation but also showed positive effects on atrophic scarring. The impact of the QS Nd:YAG laser on conditions such as acne scars and hypertrophic scars has been substantiated through various studies [9,13,16]. This laser's ability to induce thermal dermal injury while preserving the epidermis can stimulate new collagen synthesis, promote collagen remodeling, and lead to a histological thickening of the papillary dermal collagen, all with minimal downtime [13,16].

In our patient's case, there was a noticeable improvement in scarring within the four months following the last treatment. This aligns with previous observations of progressive improvement in acne scars, as seen in studies where QS Nd:YAG laser treatment led to continued incremental enhancements at the 6-month follow-up, likely attributed to ongoing collagen remodeling [13].

## 4. Conclusions

This case report underscores the incomplete understanding of the origin of hyperpigmentation following laser treatment for leg telangiectasias, underscoring the necessity for more comprehensive histopathological investigations to elucidate this pigmentation phenomenon. Correlations with post-sclerotherapy hyperpigmentation suggest hemosiderin as a potential causative pigment. While bleaching creams may effectively address pure post-inflammatory hyperpigmentation, short-pulsed lasers are advantageous when hemosiderin is anticipated.

Opting for longer wavelengths, such as the use of QS Nd:YAG 1064 nm laser, offers distinct advantages, including enhanced penetration and a reduced risk of post-inflammatory hyperpigmentation. Furthermore, it influences collagen synthesis and remodeling, which adds to its therapeutic profile. When the pigment's origin after laser therapy for leg telangiectasias is uncertain, particularly when accompanied by scarring, the QS Nd:YAG laser presents as an effective and secure treatment alternative. Larger studies are needed to assess both the efficacy and safety of the QS Nd:YAG 1064 nm laser in the treatment of skin complications, including hyperpigmentation and scars, subsequent to sclerotherapy, whether employed as an independent intervention or in combination with topical therapy.

**Author Contributions:** Conceptualization, K.H., S.B., S.M.S.J. and M.A.A.; methodology, K.H., M.A.A., L.F. and N.Y.; investigation, M.A.A. and K.H. All authors have read and agreed to the published version of the manuscript.

**Funding:** This research received no external funding.

**Institutional Review Board Statement:** Not applicable.

**Informed Consent Statement:** The patient in this manuscript has given oral and written informed consent to publication of the case details (pictures and medical data).

**Data Availability Statement:** The datasets presented in this article are not readily available because of ethical/privacy restrictions. Requests to access the datasets should be directed to the corresponding author.

**Conflicts of Interest:** The authors declare no conflict of interest.

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
