# Peer review of "Quality-Switched Nd:YAG 1064 nm Laser for Management of Hyperpigmentation and Atrophic Scars after Long-Pulsed Nd:YAG Laser for Treatment of Leg Telangiectasias—A Case Report"

_cosmetics, doi:10.3390/cosmetics10060147_

Round 1

Reviewer 1 Report

Comments and Suggestions for Authors

The authors present a case where postinflammatory changes (scarring, hyperpigmentation and/or hemosiderin) after sclerotherapy is treated with a longer wavelength QS Nd:YAG argumenting for their choice.

The case is well-presented and meticulously followed up with images of high quality.

The Discussion gives a sound overview of choices for treatment

This reviewer could not detect anything in the paper needing improvement or changes.

Author Response

R: We thank the reviewer very much for his positive and kind feedback

Reviewer 2 Report

Comments and Suggestions for Authors

The manuscript demonstrated the effectiveness of QS Nd:YAG 1064 nm Laser for the management of complications after sclerotherapy in one patient. I recommend to include in the title the type of article, a case report. 

In the conclusions section you should state that more studies are needed to assess the efficacy of QS Nd:YAG 1064 nm Laser for the management of skin complications after sclerotherapy in combination with topical therapy.

Author Response

We thank you for your valuable comments and have addressed each of your comments to the best of our knowledge.

The manuscript demonstrated the effectiveness of QS Nd:YAG 1064 nm Laser for the management of complications after sclerotherapy in one patient. I recommend to include in the title the type of article, a case report. 

R: We thank the reviewer for this suggestions and adapted the title to: QS Nd:YAG 1064 nm laser for management of hyperpigmentation and atrophic scars after long-pulsed Nd:YAG laser treatment of leg telangiectasias – a case report

In the conclusions section you should state that more studies are needed to assess the efficacy of QS Nd:YAG 1064 nm Laser for the management of skin complications after sclerotherapy in combination with topical therapy.

R: We thank the reviewer for this important comment to improve our manuscript. We have completed the conclusion section with the statement: "Larger studies are needed to assess both the efficacy and safety of the QS Nd:YAG 1064 nm laser in the treatment of skin complications, including hyperpigmentation and scars, subsequent to sclerotherapy, whether employed as independent interventions or in combination with topical therapy."